# SARS-CoV-2 and other respiratory pathogens are detected in continuous air samples from congregate settings

Mitchell D. Ramuta [1], Christina M. Newman[1], Savannah F. Brakefield[1], Miranda R. Stauss[2], Roger W. Wiseman[1,2], Amanda Kita-Yarbro[3], Eli J. O'Connor[4], Neeti Dahal[5], Ailam Lim[5], Keith P. Poulsen[5], Nasia Safdar[6], John A. Marx[6], Molly A. Accola[6], William M. Rehrauer[1,6], Julia A. Zimmer [7], Manjeet Khubbar[7], Lucas J. Beversdorf[7], Emma C. Boehm[8], David Castañeda[8], Clayton Rushford[9], Devon A. Gregory[9], Joseph D. Yao [10], Sanjib Bhattacharyya[7], Marc C. Johnson[9], Matthew T. Aliota [8], Thomas C. Friedrich [11], David H. O'Connor [1,2,12] & Shelby L. O'Connor [1,2,12] ✉

Two years after the emergence of SARS-CoV-2, there is still a need for better ways to assess the risk of transmission in congregate spaces. We deployed active air samplers to monitor the presence of SARS-CoV-2 in real-world settings across communities in the Upper Midwestern states of Wisconsin and Minnesota. Over 29 weeks, we collected 527 air samples from 15 congregate settings. We detected 106 samples that were positive for SARS-CoV-2 viral RNA, demonstrating that SARS-CoV-2 can be detected in continuous air samples collected from a variety of real-world settings. We expanded the utility of air surveillance to test for 40 other respiratory pathogens. Surveillance data revealed differences in timing and location of SARS-CoV-2 and influenza A virus detection. In addition, we obtained SARS-CoV-2 genome sequences from air samples to identify variant lineages. Collectively, this shows air sampling is a scalable, high throughput surveillance tool that could be used in conjunction with other methods for detecting respiratory pathogens in congregate settings.

Viral testing and surveillance have been a challenge throughout the COVID-19 pandemic in the United States. To date, nasal swab-based testing has predominated. Such testing did not reliably and consistently detect the first wave of SARS-CoV-2 in the United States. First-generation PCR assays were problematic and slow to deploy[1]. A massive increase in the need for PCR testing strained supply chains and laboratory capacity, leading to lengthy turnaround times[2,3]. The development of saliva-based SARS-CoV-2 testing in 2020 reduced the impact of certain supply chain bottlenecks, but laboratory capacity and test availability remained a problem in the United States[4]. Lower-

[1]Department of Pathology and Laboratory Medicine, University of Wisconsin-Madison, Madison, WI, USA. [2]Wisconsin National Primate Research Center, Madison, WI, USA. [3]Public Health Madison & Dane County, Madison, WI, USA. [4]EAGLE School, Fitchburg, WI, USA. [5]Wisconsin Veterinary Diagnostic Laboratory, Madison, WI, USA. [6]University of Wisconsin Hospitals and Clinics, Madison, WI, USA. [7]City of Milwaukee Health Department Laboratory, Milwaukee, WI, USA. [8]Department of Veterinary and Biomedical Sciences, University of Minnesota, Twin Cities, Minneapolis, MN, USA. [9]Department of Molecular Microbiology and Immunology, University of Missouri, School of Medicine, Columbia, MO, USA. [10]Department of Laboratory Medicine and Pathology, Mayo Clinic, Rochester, MN, USA. [11]Department of Pathobiological Sciences, University of Wisconsin-Madison, Madison, WI, USA. [12]These authors contributed equally: David H. O'Connor, Shelby L. O'Connor. ✉ e-mail: slfeinberg@wisc.edu

cost, point-of-care antigen tests became available in late 2020 but were not widely used for at-home testing until the arrival of the Omicron Variant of Concern in late 2021. On January 10, 2022, the United States reported the highest single-day COVID-19 case number of over 1.3 million new cases and had a seven-day case average three times greater than the previous highest peak recorded in January 2021[5,6]. Once again, testing laboratories could not scale to meet surging demand and at-home antigen test results, which are rarely reported to public health authorities, were not considered in case counts. As a result, existing testing systems for COVID-19 have provided case counts that are, at best, crude estimates for disease burden and transmission risk. Highlighting this discordance, serological data estimates that for every diagnosed case of COVID-19, there are 4.8 undiagnosed SARS-CoV-2 infections[7].

Such swab-based estimates of community infection rates are likely to become less accurate as mass testing sites start to close around the United States[8]. Antigen testing will cause fewer people to seek out formal diagnostic testing from providers who report test data to public health authorities; indeed, the United States federal government recently began distributing 500 million antigen tests without requiring mandatory results reporting[9]. There has also been a growing concern that the mental and physical exhaustion caused by COVID-19, often referred to as "pandemic fatigue," could reduce willingness to seek out testing when symptomatic, particularly if a positive test result is linked to undesirable outcomes such as mandatory isolation[10].

Accurate estimates of SARS-CoV-2 risk are especially important in congregate settings where individuals with varying degrees of risk are in close contact. Highlighting the importance of these settings, the United States invested more than $12 billion in March 2021 to expand testing in schools, workplaces, long-term care facilities, and underserved congregate settings[11]. Evidence of increased SARS-CoV-2 risk from testing programs can be used as an impetus to intensify mitigation measures, such as recommending or requiring facial masking or increased testing. Conversely, such measures can be relaxed when the risk of infection is low. This has led some to advocate for frequent, routine testing of everyone in congregate settings[12–15]. Many different approaches to high throughput testing have been deployed in support of such comprehensive testing[4,16–18], but these are expensive and difficult to scale and maintain. There is also a risk that such resource-intensive testing programs will perpetuate inequalities in the distribution of COVID-19 testing access[19–24].

Alternative "environmental" testing strategies that do not rely on individualized testing could provide a more accurate, rapid, and efficient assessment of SARS-CoV-2 infection risk in congregate settings. To date, wastewater-based surveillance for SARS-CoV-2 has received the most attention as an environmental testing strategy[25–30]. Viral RNA is shed in the feces of 30-66% of individuals with COVID-19, regardless of their symptoms[31–33], allowing SARS-CoV-2 to be detected in wastewater samples. Untreated wastewater collected at municipal wastewater treatment plants includes fecal and liquid waste from households in a sewershed and represents an efficient pooled sample that can provide information on asymptomatic and symptomatic SARS-CoV-2 infections.

Wastewater-based surveillance can provide population-wide data for large geographic regions but mainly relies on fixed sampling locations, limiting its ability to provide spatial resolution within a sewershed. Collecting sewage from individual buildings is possible, but sample collection can be challenging due to differences in the design and complexity of wastewater infrastructure. Furthermore, wastewater surveillance that relies on sample collection from wastewater treatment plants does not capture communities with decentralized systems (e.g., septic tanks) or sites where sewage is pre-treated for decontamination before reaching the wastewater treatment plant (e.g., hospitals). This is a limitation because the prevalence of COVID-19 and risk of transmission may vary substantially across different congregate settings in a community. There is still a need to develop agile and mobile surveillance technologies to collect hyperlocal data with higher resolution than is possible with wastewater.

Air surveillance is an alternative form of environmental sampling for SARS-CoV-2. Passive and active air sampling techniques have been used for air surveillance of viruses, bacteria, and fungi that are released in respiratory droplets and aerosols when infected individuals talk, cough, sneeze, and breathe[34–37]. Notably, the United States Department of Homeland Security established the BioWatch Program in 2003 to use active air samplers as routine environmental monitoring systems to detect specific biological threats to combat bioterrorism[38].

Continuous air sampling has key advantages over widespread individual testing and wastewater testing for surveillance in congregate settings. Air samples contain a mixture of exhaled components from many individuals and can capture pathogen-containing droplets and aerosols from infectious individuals, enabling virus detection independent of symptoms, test-seeking behavior, and access to swab-based testing. In contrast to wastewater surveillance, active air samplers can be easily moved to different locations, making it possible to collect surveillance data with ultrahigh spatial resolution (e.g., a single room in a building).

Several studies have shown the utility of active air samplers to detect aerosols containing SARS-CoV-2[39–44] in controlled settings and locations with known SARS-CoV-2 cases. Horve et al. demonstrated consistent detection of heat-inactivated SARS-CoV-2 virus at an aerosol concentration of 0.089 genome copies per liter of air (gc/L) when air samples were collected in a room-scale experiment during an 8-h interval[39]. Another study compared the effectiveness of surface and bioaerosol sampling methods to detect SARS-CoV-2 and showed active air samples detected SARS-CoV-2 in 53% of the samples when run for 1–2 h in hospital rooms of COVID-19 patients, while passive air sampling and surface swabs detected SARS-CoV-2 in only 12% and 14% of samples, respectively[40]. Parhizkar et al. used active air samplers to assess the relationship between COVID-19 patient viral load and environmental viral load in a controlled chamber over three days. Increases in patient viral load were associated with lower cycle threshold (Ct) values detected in near (1.2 m) and far (3.5 m) air samplers[41]. Lastly, a study demonstrated the utility of using active air samplers to track the presence and concentration of virus in air longitudinally during COVID-19 isolation periods in student dormitories. The study observed a significant increase in Ct values for COVID-19 positive students after their first test, as well as in environmental samples as individuals recovered indicating a reduction in virus presence[42]. These studies provide proof of concept on the feasibility of using active air samplers to detect SARS-CoV-2. However, each study was performed in a controlled environment occupied by COVID-19 positive individuals.

Here, we evaluate whether active air samplers can be used for prospective air surveillance of SARS-CoV-2 in real-world congregate settings, where pathogen-containing aerosols are likely present at a much lower concentration, and the presence of positive individuals is unknown. This study addresses a key knowledge gap in how active air samplers perform as routine pathogen monitoring systems in real-world settings. We demonstrate that it is feasible to use active air samplers for air respiratory pathogen detection and sequencing across different types of congregate settings.

## Results
### Study design
From July 19, 2021, to February 9, 2022, continuous air samplers were deployed in several public locations to survey SARS-CoV-2 in the environment of real-world settings. We used Thermo Fisher Scientific AerosolSense Samplers for daily and weekly air surveillance at places considered to be high-risk for close-contact, indoor SARS-CoV-2 transmission. Air cartridges were collected and tested for SARS-CoV-2

**Table 1 | SARS-CoV-2 air sample results**

| Location | Site Name | Start Date | End Date | Number of Samples | Positive | Negative | Inconclusive |
|---|---|---|---|---|---|---|---|
| Dane County, WI | Preschool #1 | 08.18.2021 | 02.08.2022 | 49 | 3 | 43 | 3 |
| | Preschool #2 | 08.11.2021 | 10.14.2021 | 22 | 2 | 18 | 2 |
| | School #1 | 07.26.2021 | 02.08.2022 | 73 | 4 | 62 | 7 |
| | School #2 | 10.14.2021 | 02.09.2022 | 15 | 8 | 5 | 2 |
| | School #3 | 12.14.2021 | 02.08.2022 | 7 | 7 | 0 | 0 |
| | School #4 | 12.15.2021 | 02.08.2022 | 8 | 6 | 1 | 1 |
| | Hospital | 08.20.2021 | 10.25.2021 | 51 | 18 | 33 | 0 |
| | Campus Athletic Facility | 07.19.2021 | 02.09.2022 | 179 | 20 | 141 | 18 |
| | Campus Coffee Shop | 08.17.2021 | 02.03.2022 | 54 | 5 | 44 | 5 |
| | Office | 09.30.2021 | 12.10.2021 | 8 | 0 | 8 | 0 |
| Minneapolis, MN | Brewery taproom | 10.18.2021 | 02.7.2022 | 26 | 11 | 2 | 13 |
| Rochester, MN | Bar | 09.27.2021 | 11.24.2021 | 9 | 5 | 4 | 0 |
| | Hospital Cafeteria | 09.20.2021 | 11.24.2021 | 10 | 6 | 4 | 0 |
| Milwaukee, WI | Emergency Housing Facility #1 | 12.17.2021 | 02.08.2022 | 9 | 5 | 3 | 1 |
| | Emergency Housing Facility #2 | 12.17.2021 | 02.08.2022 | 7 | 6 | 1 | 0 |
| | | | Total | 527 | 106 | 369 | 52 |

Dates are listed as MM.DD.YYYY.

RNA by quantitative reverse transcription PCR (RT-qPCR) and transcription-mediated amplification (TMA) (Supplementary Fig. 1). Several different RT-qPCR and TMA assays were used to test samples for viral RNA (vRNA) throughout the study, depending on the location of the test site and the availability of testing at the time. Because the PCR assays were different, Ct values cannot be directly compared between testing sites. Further details on the cut-off values used for calling air samples positive, inconclusive, or negative are described in the methods section and Supplementary Data 1. We developed a user-friendly workflow to collect air cartridge metadata, upload test data, and report results to surveillance sites within 24–48 h of receiving air cartridges for testing (Supplementary Fig. 1), enabling non-technical staff (e.g., custodial staff, students) to exchange and catalog cartridges easily and accurately.

## SARS-CoV-2 detection in community settings

To demonstrate the utility of air surveillance in real-world settings, we chose a diversity of community locations for placement: a campus coffee shop, hospital, office, campus athletic training facility, brewery taproom, cafeteria, bar, two preschools, four K-12 schools, and two shelters located throughout Wisconsin and Minnesota. We collected 527 air cartridges from the 15 locations to test for the presence of SARS-CoV-2 viral RNA (Table 1; Supplementary Data 1). Four hundred sixty-six (88.4%) air samples were collected from testing sites in Dane County, Wisconsin, 26 (4.9%) from Minneapolis, Minnesota, 19 (3.6%) from Rochester, Minnesota, and 16 (3.1%) from Milwaukee, Wisconsin. During the 29 weeks, Dane County experienced a moderate-to-high transmission rate of COVID-19 cases despite having a high county-wide vaccination rate (>65% adults with two doses) (Supplementary Fig. 2). Increases in COVID-19 cases were observed following the emergence of Delta and Omicron Variants of Concern in the community. Public Health Madison and Dane County instituted an emergency mask mandate on August 19th, 2021, that was extended throughout the entire study. The order required every individual aged two and older to wear a face-covering in most public enclosed spaces, including K-12 schools. Exceptions were made in spaces where all people were known to be vaccinated.

Throughout the study, we detected a total of 106 SARS-CoV-2 positive and 52 inconclusive air cartridges (an inconclusive result is defined when at least one of the PCR targets is positive while at least one of the PCR targets is negative). We were able to intermittently identify SARS-CoV-2 positive air cartridges at 14 of the 15 surveillance sites (Table 1), even when intensive risk mitigation strategies were recommended by public health.

We did not perform intensive, routine SARS-CoV-2 swab-based testing on all individuals in these real-world settings during the study, so it was impossible to know the SARS-CoV-2 status of every person who spent time in the proximity of the samplers. However, we were able to retrospectively correlate air surveillance data with reported cases during a prolonged COVID-19 outbreak at one of the testing sites (Fig. 1). Air samples were routinely collected and tested before, during, and after the outbreak, which resulted in a total of 20 confirmed cases. Five of the confirmed cases were reported by individuals who were either symptomatic or tested positive for SARS-CoV-2 while working on-site at the congregate setting. People who congregated in the same rooms as these five individuals were considered to be close contacts and were quarantined at home after the case was reported. Fifteen of the confirmed cases were reported by individuals during their at-home quarantine. Air samples were positive before the first documented case of COVID-19 and throughout the outbreak. SARS-CoV-2 RNA was detected in an air sample for the first time between days five and seven, preceding the first confirmed case by seven days. Air samples collected after day 23 were either inconclusive or negative for SARS-CoV-2. No reported cases were observed in the building during this time. No air sample was collected from days 28–30 because the air cartridge was inserted improperly, leading to a machine error. It should be noted that, at the time of this outbreak, we did not have sufficient data on air sampling accuracy to make recommendations to the affected congregate setting to intensify risk mitigation.

## Extending the duration of continuous air sampling

Daily air surveillance programs are resource-intensive and expensive. To reduce the cost and complexity of surveillance programs, we tested the feasibility of extending the sampling interval while maintaining the detection sensitivity of daily testing. Over the course of five weeks, two adjacent AerosolSense instruments were deployed to either run continuously (~168 h) or daily (~24 h) over several days. Air samples were gathered throughout the week, nucleic acids were isolated simultaneously, and tested with two RT-qPCR Center for Disease Control and Prevention (CDC) assays targeting SARS-CoV-2 N1 and N2. During the first week of sampling, both air samplers detected SARS-CoV-2 RNA, showing continuous air sampling for a week captured similar copies of

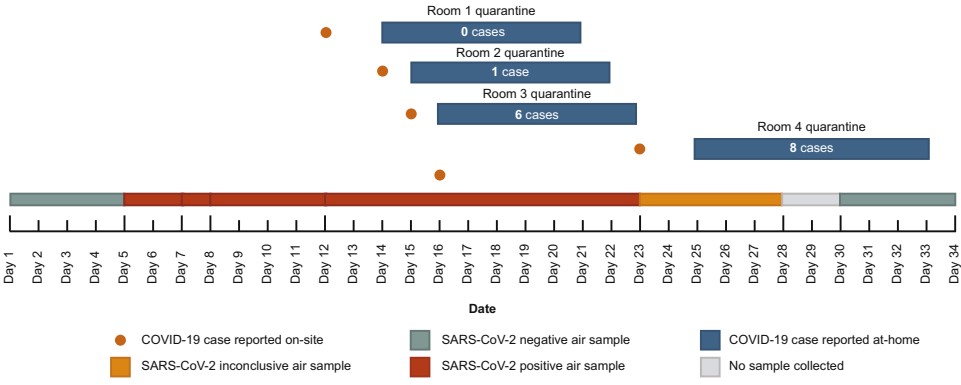

**Fig. 1 | COVID-19 outbreak timeline.** Confirmed COVID-19 cases and air sample SARS-CoV-2 RT-qPCR results in the congregate setting are plotted over time. Orange dots represent confirmed COVID-19 cases from individuals present in the building. Blue boxes show the number of COVID-19 cases that occurred while close contacts were in quarantine. Air sample SARS-CoV-2 RT-qPCR results are represented by boxes as positive (red), negative (green), or inconclusive (orange). The gray box indicates that no sample was collected during that time period.

viral RNA, determined by Ct values, when compared to more resource-intensive daily sampling (Fig. 2). Weekly and daily air sample results were concordant for the next two weeks; air samples were either negative or inconclusive. Continuous air samples were negative for the last two weeks, while daily sampling identified one inconclusive sample during each of these weeks. However, each of the inconclusive samples had N1 Ct values >40 and high N2 Ct values of 39.1 and 39.4, respectively. This suggests that congregate settings can use either daily or weekly sampling to balance cost and turnaround time while maintaining detection sensitivity.

### Expanding detection to additional respiratory pathogens

To explore whether pathogens other than SARS-CoV-2 are detected in the same collected air sample, we tested them for the presence of nucleic acids of 40 other viral, bacterial, and fungal respiratory pathogens using the TrueMark Respiratory Panel 2.0 Array Card on a QuantStudio 7 Pro Instrument[45]. We assessed the LOD for each TrueMark assay using contrived air samples to determine pathogen-specific thresholds for calling samples positive or negative. Contrived air samples were created and tested in quadruplicate by spiking the TrueMark Respiratory Panel 2.0 Amplification Control (Thermo Fisher Scientific) plasmid into pooled air samples at concentrations of 0, 1.25, 12.5, 50, and 250 copies per reaction. Air samples were collected for 48 h from an empty office to minimize any background pathogens present in the samples (see methods section for more details). Pathogen-specific thresholds were determined by averaging the cycle relative threshold (Crt) values of positive replicates at the lowest dilution concentration with at least 75% positive replicates. Cycle relative threshold values listed in Supplementary Table 1 were used as cut-off values for positivity for the detection of each pathogen.

From October 25, 2021, to February 9, 2022, air samples were collected weekly from eight sites in Dane County, Wisconsin: a campus coffee shop, a preschool, an office space, a campus athletic training facility, and four K-12 schools. Semi-quantitative RT-PCR was performed on 98 air samples using this TrueMark Respiratory 2.0 Panel. During the 15 weeks of air surveillance, we detected 15 different respiratory pathogens across the eight sites (Supplementary Table 2). Commensal or transiently commensal respiratory tract microbes were frequently detected in air samples at each site, including *Klebsiella pneumoniae*, *Staphylococcus aureus*, and *Moraxella catarrhalis* (Fig. 3). The panel also detected respiratory pathogens associated with illness in school-aged children, including adenovirus, human coronavirus OC43, Epstein-Barr virus, cytomegalovirus, influenza A virus, parainfluenza virus 3, respiratory syncytial virus A, and SARS-CoV-2. Certain

pathogens, such as human bocavirus, were consistently detected only in settings where there were young children, consistent with its widespread distribution in this population, highlighting that different types of congregate settings have distinctive air surveillance pathogen signatures (Fig. 3b)[46].

The pattern of influenza A virus (IAV) nucleic acid detection was especially striking. The Wisconsin Department of Health Services issued a health alert on November 30, 2021, noting increased IAV activity among college and university students[47]. As shown in Fig. 4, we detected IAV nucleic acid in the air collected from two AerosolSense samplers deployed on the University of Wisconsin-Madison campus beginning the week of November 10, 2021. Air collected from both of these samplers contained IAV nucleic acids from mid-November 2021 to January 2022. One sampler, located in a campus coffee shop, was negative for IAV beginning the week of December 22, 2021, through the week ending January 12, 2022 (Fig. 4a); this coincided with the end of the UW-Madison fall academic semester and the beginning of holiday break. During this time, the coffee shop was closed to customers from December 17, 2021, to January 18, 2022, but the building was still open for repairs and cleaning. We continued to detect IAV nucleic acids collected by the sampler in the training facility as student-athletes and staff remained on campus for training and competition during the holiday break. Strikingly, IAV nucleic acids were only sporadically detected in air samplers located on non-campus community sites in Dane County, Wisconsin. In contrast, SARS-CoV-2 nucleic acids were frequently detected at testing sites across the community (Fig. 4b). Dane County experienced a high transmission rate of COVID-19 during this time, and the detection of SARS-CoV-2 positive air samples increased at testing sites following the emergence of Omicron. Overall the differential detection of IAV and SARS-CoV-2 nucleic acids is consistent with the localized known IAV campus outbreak and widespread SARS-CoV-2 transmission.

### Sequencing of SARS-CoV-2 RNA from collected air samples

Throughout the pandemic, deep sequencing of SARS-CoV-2 RNA extracted from clinical samples and wastewater has played a crucial role in monitoring viral evolution and tracking variants of concern. We used two sequencing strategies to obtain partial and near-full genome SARS-CoV-2 sequences from 11 air samples to demonstrate the feasibility of genotyping SARS-CoV-2 from collected air. Sequencing efforts were focused on air samples with low Ct values from congregate settings where individuals often removed their masks to eat and drink (e.g., taprooms, bars, cafeterias, and shelters).

Targeted sequencing was performed on nine samples collected from two AerosolSense samplers in a brewery taproom in Minnesota

| Sampler/Day | Day 1 | Day 2 | Day 3 | Day 4 | Day 5 | Day 6 | Day 7 |
|---|---|---|---|---|---|---|---|
| Continuous Wk 1 | 35.4 \| 36.5 | | | | | | |
| Daily Wk 1 | Neg | 37.4 \| 37.1 | Neg | Neg | Neg | Neg | Neg |
| Continuous Wk 2 | 36.4 \| Undet. | | | | | | |
| Daily Wk 2 | 36.6 \| Undet. | | Neg | Neg | Neg | Neg | Neg |
| Continuous Wk 3 | Neg | | | | | | |
| Daily Wk 3 | Neg | | Neg | | Neg | Neg | Neg |
| Continuous Wk 4 | Neg | | | | | | |
| Daily Wk 4 | Undet. \| 39.1 | Neg | Neg | Neg | Neg | Neg | Neg |
| Continuous Wk 5 | Neg | | | | | | |
| Daily Wk 5 | Neg | Neg | Neg | Neg | Neg | Neg | Undet. \| 39.4 |

**Fig. 2 | Comparison of SARS-CoV-2 RT-qPCR results from continuous and daily air sampling intervals.** Two adjacent Thermo Scientific AerosolSense instruments were run continuously or daily over several days. SARS-CoV-2 genomic material was detected by two separate RT-qPCR CDC assays. If the cycle threshold values of one or both N1 and N2 assays were less than 40, the Ct values are shown in the box separated by a "|". "Undet." was used for assays that had Ct values greater than 40. If the Ct values of both RT-qPCR assays were greater than 40, the boxes are labeled as "Neg". Samples considered to be positive are shaded red. Boxes shaded in gray are either inconclusive or negative, marked "Neg".

between November 22, 2021, and January 25, 2022, using primers targeting the SARS-CoV-2 spike gene receptor-binding domain (RBD)[27]. Consensus sequences from four samples collected between November 22 and December 13, 2021, all contained the characteristic S:L452R and S:T478K variants associated with the Delta lineage that predominated at this time (Table 2). Interestingly, one cartridge collected viral RNA that had two rare consensus variants, S:F456L and synonymous S:F562F, suggesting that sequencing can detect unexpected variants in air samples and could be used for detecting newly emerging variants of concern in congregate settings.

The brewery taproom implemented a vaccine mandate on December 10, 2021, after observing an increase in SARS-CoV-2 detection in air samples and COVID-19 cases in the community. The vaccine mandate required customers to show proof that they received all recommended doses in their primary series of COVID-19 vaccines or had a negative COVID-19 test within the last 72 h for indoor dining. No air samples collected between December 13, 2021, and December 30th, 2021, were positive for SARS-CoV-2. However, SARS-CoV-2 RNA was detected in 72% of the air samples collected following the emergence of Omicron. Six air samples (N1 Ct values <36) collected between December 30, 2021, and January 25, 2022, all contained characteristic S:K417N, S:N440K, S:G446S, S:S477N, S:T478K, S:E484A, S:Q493R, S:G496S, S:Q498R, S:N501Y, S:Y505H, S:T547K variants associated with the Omicron BA.1 lineage coinciding with the emergence of Omicron in the region (Table 2). In early January, we detected both Delta and Omicron sequences in one of the air samples. These data support that virus genetic material collected by air samples parallels the longitudinal detection of the same lineages transmitted in the community.

The remaining two additional samples with Ct values below 32 collected from two shelters in Milwaukee, Wisconsin, from December 21, 2021, to January 7, 2022, were examined using the ARTIC protocol followed by Illumina sequencing. Sequence coverage was incomplete for both samples, with 28% and 4% of the sequences having low coverage or missing data. However, there was enough information to assign both samples to the Omicron BA.1 lineage (Table 2).

## Discussion

An increasing number of public health organizations have employed environmental surveillance methods, in conjunction with individual clinical testing, to provide data on SARS-CoV-2 prevalence, monitor viral evolution, and track viral variants in communities[25,30,48,49]. Environmental surveillance tools could help public health make data-driven decisions for implementing COVID-19 interventions and allocating resources within a community. This study demonstrates the feasibility of using active air samplers for environmental pathogen surveillance in real-world settings. Air surveillance may improve our ability to identify pathogen-containing aerosols present in congregate settings to assess the risk of transmission with high spatial resolution, providing a more complete picture to public health officials.

Air sampling has several key advantages that make it an especially attractive surveillance strategy. Air surveillance bypasses test-seeking behavior and can be used to monitor multiple individuals and pathogens in the same sample. Additionally, task-shifting air sampling cartridge management to individuals with no scientific training makes air sampling more accessible and scalable than individual testing with specialized personnel. Deploying networks of air samplers in congregate settings could be a non-invasive way of assessing the risk of transmission and evaluating risk mitigation strategies at the level of a single congregate space (e.g., a single classroom, bar, or emergency shelter), while also improving public health awareness of pathogen circulation within a community on a large-scale. For example, in nursing homes, where more than 150,000 residents have died of COVID-19, detection of SARS-CoV-2 in air samples could be used as a trigger for intensifying risk mitigation efforts to protect residents[50,51].

Public and private schools are particularly appealing targets for establishing air surveillance networks because they are geographically distributed throughout each state, such that air sampling results in schools could be generally representative of the communities in which they are located. There are 5987 public and private elementary and secondary schools in Wisconsin, plus 2716 preschool and child development centers[52,53]. At ~$5000 per air sampler (https://www.thermofisher.com/order/catalog/product/AEROSOLSENSE), each school in the state could be equipped with

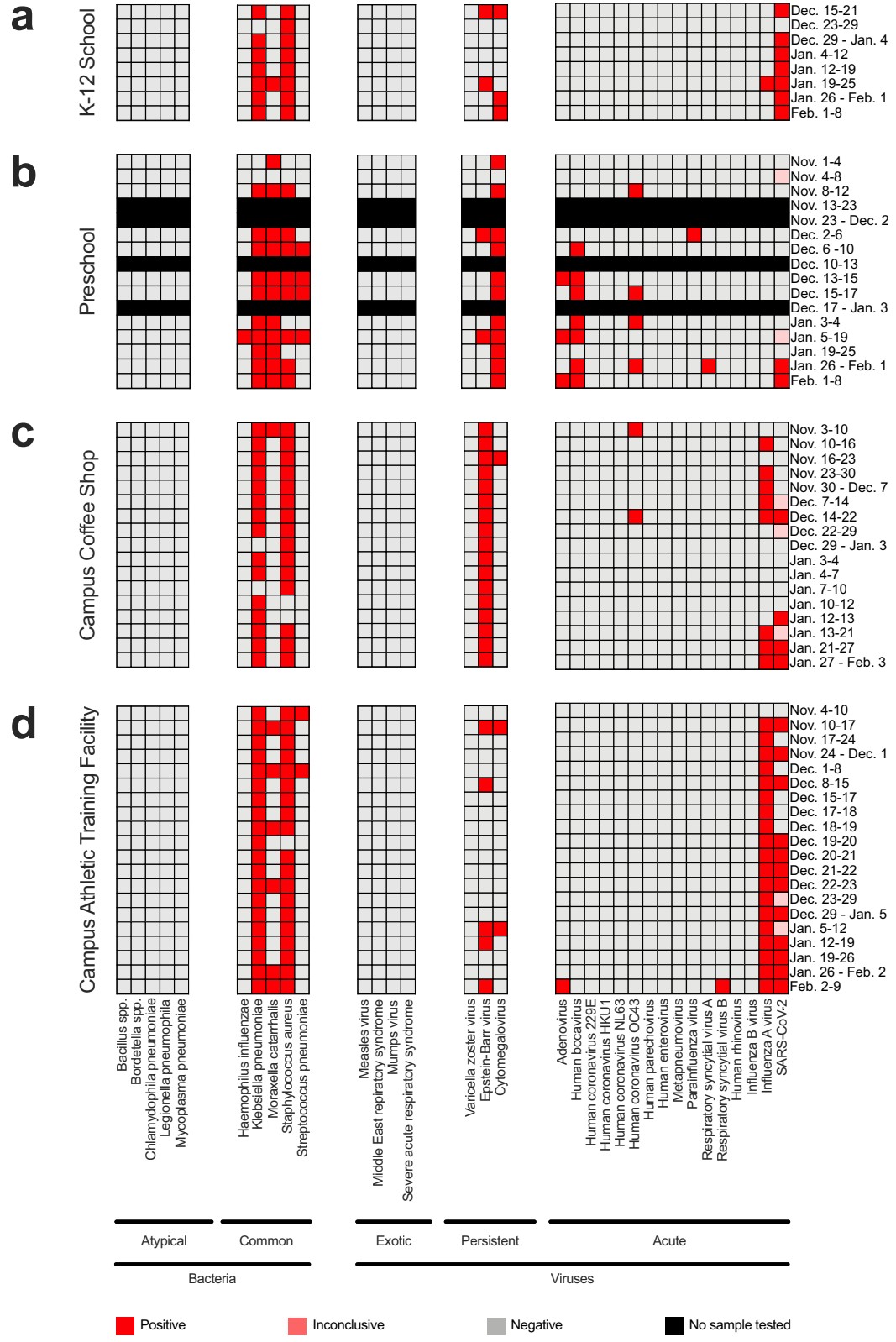

**Fig. 3 | In-air respiratory pathogen detection in congregate settings.**
**a** Respiratory pathogen detection in air samples collected from a K-12 school,
**b** preschool, **c** campus coffee shop, and **d** campus athletic facility. Genomic
material from 40 respiratory pathogens was detected by semi-quantitative RT-PCR
using the TrueMark Respiratory 2.0 TaqMan Array Card. SARS-CoV-2 genomic
material was detected by two separate RT-qPCR CDC assays. Boxes shaded in red,
pink, and gray represent positive, inconclusive, and negative air samples collected
during the sampling interval on the y-axis. No sample was tested for boxes shaded
in black.

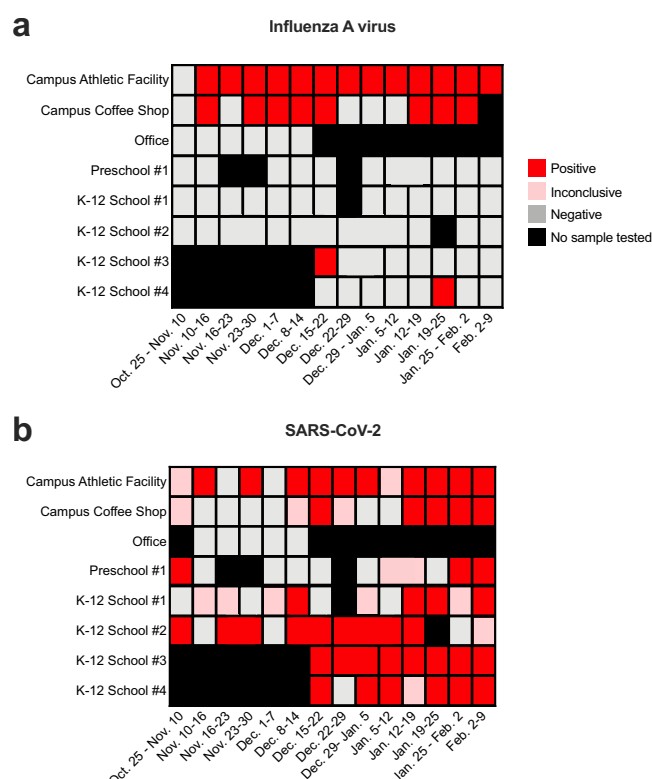

**Fig. 4 | Detection of SARS-CoV-2 and Influenza A virus in Dane County, WI.**
**a** Influenza A virus (IAV) detection in air samples collected from congregate set-tings. IAV genomic material was detected by semi-quantitative RT-PCR using the TrueMark Respiratory 2.0 TaqMan Array Card. **b** SARS-CoV-2 detection in air samples collected from congregate settings. SARS-CoV-2 genomic material was detected by two separate RT-qPCR N1 and N2 CDC assays. Boxes shaded in red, pink, and gray represent positive, inconclusive, and negative air samples collected during the sampling interval in the x-axis. No sample was tested for boxes shaded in black. Campus sites were located on the college campus of the University of Wisconsin-Madison.

an air sampler for ~$43.5 million. Assuming daily testing for 180 school days per academic year at a cost of $115 per sample ($40 per cartridge and $75 per test), a testing program would cost about $180 million[54]. Substituting low-cost nylon flocking material for the AerosolSense Capture Material, as described by others, could enable cartridge recycling, substantially reducing both cost and plastic waste, which would make routine, long-term air surveillance in such a large number of settings more feasible[39].

Constructing individualized air sampling workflows that are both actionable and cost-effective for each specific setting will require fur-ther investigation. Weekly sampling may be more appropriate when the concerns of respiratory pathogens are low, while daily sampling may be more appropriate during periods of high transmission (ie. mid-November to February). Development of easy and effective rapid on-site testing tools that can be used at the point-of-collection could also improve the value of air sampling in certain locations when a more rapid turnaround time has clear advantages. We chose to focus our study on weekly air sampling to assess both the scientific and social feasibility of an air sampling program. We captured circulating viruses at several community partner sites that relied on the cooperation of staff members with a diversity of backgrounds to manage their on-site instrument. This approach demonstrated that testing air samples at weekly intervals can capture similar results as daily sampling, reduce the required amount of testing resources, lessen the overall workload, and provide valuable data on what pathogens are circulating at a given time.

Networks of air samplers deployed as described here could also play a role in improving public health resilience to new and emerging respiratory diseases. Had a nationwide network existed in the United States prior to the arrival of SARS-CoV-2, the spread of the virus through space and time could have been more accurately evaluated. Moreover, adding continuous air sampling in the future at ports of entry and, potentially, aboard international aircrafts could intercept vessels and passengers harboring worrisome respiratory pathogens. Improvements in technologies that enable real-time, highly multi-plexed pathogen detection and genotyping could be leveraged with air sampling to improve quarantine effectiveness. Consider the arrival of the Omicron variant in the United States: it was initially reported to the World Health Organization by South African authorities on November 24, 2021[55]. However, the first confirmed Omicron case occurred weeks earlier in a sample collected on November 9, 2021. Additionally, con-tinuous genomic surveillance of air by targeted spike or whole-genome sequencing from international travelers arriving in the United States could have shortened the window of first detection of Omicron in this country. Establishing a network of air surveillance programs now could provide an early warning for the arrival of future SARS-CoV-2 variants, as well as future novel respiratory viruses of concern.

We used RT-PCR and TMA analyses to detect pathogen nucleic acids captured in air samples. The nucleic acid amplification tests used in this study provide semi-quantitative information on the viral load present in ambient air. Absolute quantification could provide more quantitative information on the environmental aerosol viral load[34]. The number of SARS-CoV-2 gene copies captured by an air sample can be used to calculate the concentration of gene copies per cubic meter of sampled air during the sampling interval. However, viral load data should be interpreted with care when collecting air samples from congregate settings with complex structures and ventilation systems. The viral RNA load collected by active air samplers depends on many factors, including the location of the sampler, amount of air collected, ventilation capacity, the amount of virus shed by each infected indi-vidual, the number of infected individuals in an area, and the dimen-sions of the indoor environment. Deploying a single air sampler in a congregate setting with the size and complexity of a school may not accurately reflect the presence of all pathogens in the air. Multiple air samplers may be required to more accurately detect the presence of infected individuals in a building with multiple different air compart-ments and identify SARS-CoV-2 transmission hotspots[56]. Increasing the number of air samplers deployed in a surveillance program could substantially increase the cost of a monitoring program. Sousan et al. pursued sampling air directly from a heating, ventilation, and air conditioning system of a large student dormitory as an alternative sampling method to collect an efficient sample from a complex structure[57]. Further studies are needed to determine how effective this strategy is in different congregate settings.

Additionally, detection of viral RNA by RT-qPCR analysis cannot determine whether an air cartridge is collecting infectious virus or viral RNA that does not pose a risk for infection. It remains challenging to use the PCR Ct value to predict whether SARS-CoV-2 isolated from air samples and clinical specimens will be culturable[58,59]. This may cause biases in settings where UV germicidal irradiation (UVGI) is used. Depending on the UVGI exposure time, SARS-CoV-2 can be inactivated without largely impacting the detection of genetic material by con-ventional PCR[60]. Air samples collected in these environments following UVGI treatment could be positive by PCR without the presence of infectious viruses. Nonetheless, a nucleic acid positive air sample result in these settings could indicate an infected individual was present in the space and might still pose a risk for transmission. We did not attempt to culture SARS-CoV-2 or any other pathogens from air car-tridges in this study to determine if infectious virus can be isolated from the nucleic acid positive samples. Several studies have attempted to culture SARS-CoV-2 and other viruses from air samples with mixed

**Table 2 | Air sample SARS-CoV-2 sequencing results**

| Location | Air sample barcode | Start | Finish | SARS-CoV-2 N gene PCR Ct | Lineage(s) | Spike RBD amino acid differences vs. SARS-CoV-2 reference | Bioproject Accession Number |
|---|---|---|---|---|---|---|---|
| Emergency Housing Facility #1 | AE0000010795F4A | 12.21.2021 | 01.07.2022 | 31.77 | BA.1 | S477N; T478K; E484A; Q493R; G496S; Q498R; N501Y; Y505H; T547K | PRJNA856293 |
| Emergency Housing Facility #2 | AE0000010795B42 | 12.21.2021 | 01.07.2022 | 25.94 | BA.1 | S477N; T478K; E484A; Q493R; G496S; Q498R; N501Y; Y505H; T547K | PRJNA856293 |
| Brewery taproom | AE0000010466C36 | 11.22.2021 | 11.29.2021 | 30.5 & 34.28 | Delta | L452R; T478K | PRJNA811594 |
| Brewery taproom | AE0000010464938 | 11.22.2021 | 11.29.2021 | 31.63 & 35.15 | Delta | L452R; T478K | PRJNA811594 |
| Brewery taproom | AE0000010467837 | 12.06.2021 | 12.13.2021 | 38 & 69.63 | Delta | L452R; T478K | PRJNA811594 |
| Brewery taproom | AE0000010467A3C | 12.06.2021 | 12.13.2021 | 37.41 & 42.9 | Delta | L452R; F456L; T478K; F562F | PRJNA811594 |
| Brewery taproom | AE0000010463B32 | 12.30.2021 | 01.03.2022 | 35.28 & 38.77 | BA.1 | K417N; N440K; G446S; S477N; T478K; E484A; Q493R; G496S; Q498R; N501Y; Y505H; T547K | PRJNA811594 |
| Brewery taproom | AE0000010464442E | 12.30.2021 | 01.03.2022 | 35.69 & 37.9 | Delta and BA.1 | K417N; N440K; G446S; L452R; S477N; T478K; E484A; Q493R; G496S; Q498R; N501Y; Y505H; T547K | PRJNA811594 |
| Brewery taproom | AE0000010463F3A | 01.03.2022 | 01.10.2022 | 33.85 & 37.47 | BA.1 | K417N; N440K; G446S; S477N; T478K; E484A; Q493R; G496S; Q498R; N501Y; Y505H; T547K | PRJNA811594 |
| Brewery taproom | AE0000010465530 | 01.10.2022 | 01.17.2022 | 33.37 & 36.34 | BA.1 | K417N; N440K; G446S; S477N; T478K; E484A; Q493R; G496S; Q498R; N501Y; Y505H; T547K | PRJNA811594 |
| Brewery taproom | AE0000010053FA3D | 01.17.2022 | 1/25/22 | 34.04 & 38.94 | BA.1 | K417N; N440K; G446S; S477N; T478K; E484A; Q493R; G496S; Q498R; N501Y; Y505H; T547K | PRJNA811594 |

Samples with two-cycle threshold (Ct) values listed in the table were tested with two SARS-CoV-2 N1 and N2 assays. N1 and N2 Ct values are separated by "&". Samples with one Ct value listed in the table were tested with the Applied Biosystems TaqPath™ COVID-19 Combo Kit. Accession numbers list the respective Sequence Read Archive (SRA) bioproject. Dates are listed as MM.DD.YYYY.
RBD receptor binding domain, Ct cycle threshold.

results[43,44,61–64]. Furthermore, previous studies have shown that the air sampling method, interval, and capture media can affect viral integrity[65,66]. AerosolSense samplers use an accelerating slit impactor to collect aerosol particles on dry filter capture substrate. Air sampling methods that rely on impactors and filters are not optimal for maintaining virus viability because of damage caused by impaction forces and dehydration during the collection process. Live virus recovery from continuous air samples would be valuable, as it might potentiate culture and expansion of unknown pathogens with greater sensitivity.

We did not perform intensive swab-based SARS-CoV-2 surveillance testing of individuals in congregate settings during this study to systematically evaluate how air sampling data correlates with the presence of infected individuals. However, COVID-19 case report data was generously shared with us from a congregate setting with on-site testing during a prolonged COVID-19 outbreak. These data suggest that congregate risk during this outbreak could have been estimated with air sampling data alone had individual testing not been available or widespread, and that air sampling provided an early indication of SARS-CoV-2 transmission risk. Future studies will need to be performed to evaluate further how well air surveillance data correlates with individual testing data in different settings. Such studies would require a partnership with a well-established, intensive surveillance testing program to compare air sampling data with case reports from asymptomatic and symptomatic individuals. For example, the Oregon Child Absenteeism Due to Respiratory Disease Study (ORCHARDS) enrolls students and families from a K-12 school district and the surrounding community to understand the causes of influenza-like illness[67]. Partnering with a study like ORCHARDS would provide an opportunity to explore the relationship between data collected by air surveillance and identified causes of respiratory infections in school settings.

After two years of COVID restrictions, there is pushback against public health measures to counteract virus transmission[10,68]. One component of this resistance is that guidelines issued at the national, state, or even municipal level do not necessarily reflect hyperlocal risk within specific congregate settings: an individual school, sports arena, bar, etc. Air sampling provides a measure of risk with this level of granularity. However, care must be taken when interpreting and sharing air sampling data. In some settings, stakeholders may choose to be liberal in disclosing air sampling results, sharing this information with employees, customers, visitors, and others so they can individually modulate their risk mitigation. In other settings, public health and testing laboratories may work directly with the leadership of congregate settings to couple air sampling data with action. For example, one of the county public health departments involved in this work offers enhanced testing to sites where high levels of SARS-CoV-2 are detected in the air, while a second public health department created a flow chart describing how schools might respond to positive air sampling data if there are no known cases of SARS-CoV-2 in a given school (Supplementary Fig. 3).

We did not have sequencing data available from the brewery taproom to see the transition from Delta to Omicron in real-time. In retrospect, sequencing and RT-qPCR data could have been used to help make data-driven decisions to adapt the risk mitigation strategy. Adjustments to the COVID-19 policy could have included increasing the ventilation or expanding the vaccine mandate to require booster doses that have been shown to improve protection against Omicron[42,69]. However, even with these data available, congregate settings may be hesitant to increase risk mitigation strategies past the most stringent guidelines set out by the CDC and local public health. Furthermore, some settings may have no appetite for COVID-19 risk mitigation regardless of air surveillance results. Environmental surveillance in these settings may nonetheless be valuable to public health alone, allowing them to anticipate and respond quickly to surges in respiratory disease[70]. In fact, in settings where diagnostic testing for

SARS-CoV-2 is limited by pandemic fatigue and apathy towards risk minimization measures, air sampling could be exceptionally useful in providing baseline data on respiratory virus levels that would otherwise be impossible to obtain.

Taken together, these results show that continuous air surveillance with active air samplers can unambiguously detect respiratory pathogens, including SARS-CoV-2, in congregate settings. Similar to the National Wastewater Surveillance System recently established by the US Centers for Disease Control and Prevention, expansion of air surveillance efforts could provide additional safeguards for congregate settings and improve resilience to future respiratory virus threats[25].

## Methods
### Collection of air samples
The Institutional Review Board of the University of Wisconsin-Madison Health Sciences waived ethical approval for this work. AerosolSense instruments (Thermo Fisher Scientific, cat. 2900AA) were deployed in various indoor community settings for air pathogen surveillance. Samplers were placed in high-traffic areas on flat surfaces 1-1.5 meters off the ground and calibrated to sample 200 liters of air per minute. AerosolSense cartridges (Thermo Fisher Scientific, cat. 12148001) were installed and removed from the air sampler according to the manufacturer brochure and transferred to the lab for testing[71]. We developed a workflow to simplify data collection, management, and reporting (Supplementary Fig. 1). The workflow relies on the iOS and Android Askidd mobile app to easily collect air cartridge metadata and upload it to a centralized LabKey database. Air sampler users simply open the Askidd app, and take a picture of the air cartridge barcode when installed and removed from the machine. The Askidd app collects GPS coordinates of the air sampler, timestamp, AerosolSense instrument ID, and air cartridge barcode to send to LabKey. When air sample testing was completed in the lab the results were uploaded to the Labkey database and displayed in the Askidd mobile app. This workflow tracks data for every cartridge and limits user errors that could occur during manual input.

### Detection of SARS-CoV-2
**University of Wisconsin–Madison**

### Hologic Aptima SARS-CoV-2 assay
AerosolSense cartridges collected at a hospital in Dane County, Wisconsin, from August 20th to October 25th, 2021, were tested for SARS-CoV-2 viral RNA using the Aptima SARS-CoV-2 Assay (Hologic) on the Panther System (Hologic). The Aptima SARS-CoV-2 Assay was authorized for emergency use authorization (EUA) by the United States Food and Drug Administration (FDA) for the qualitative detection of vRNA[72]. Air cartridges were collected from AerosolSense Samplers as recommended by the manufacturer. One substrate was removed from the cartridge using sterile forceps, transferred to a tube containing 750 μL of universal transport medium (Copan), and incubated at room temperature for 5–10 min. Following the incubation, 500 μL of the eluate was transferred to a Panther Fusion Specimen Lysis Tube (Hologic) containing 710 μL of specimen transport medium. The tube was gently mixed by inverting it several times before loading it into the Panther System to automatically run the Aptima SARS-CoV-2 Assay as described by the manufacturer. Aptima SARS-CoV-2 positive and negative controls were run with each set of air samples. According to the manufacturer's recommendations, an estimated cut-off value of >650 RLU was used to consider samples as SARS-CoV-2 positive.

### CDC SARS-CoV-2 RT-qPCR assay
RNA extraction and real-time reverse-transcription polymerase chain reaction (RT-qPCR) testing of the air cartridge specimens occurred at the University of Wisconsin-Madison WVDL-WSLH COVID Laboratory

(WWCL, Madison, Wisconsin). Each air cartridge was submerged in 500 μL of 1X phosphate-buffered saline (PBS) for at least 1 h. For all air cartridges, the tubes were vortexed vigorously and 190 μL of the PBS was used for RNA extraction using the MagMAX Viral/Pathogen II (MVP II) Nucleic Acid Isolation Kit (Thermo Fisher Scientific) on a 96-well King-Fisher Flex extraction platform and eluted in a volume of 50 μL according to manufacturer's instructions. A multiplex one-step RT-qPCR assay targeting the 2019-nCoV N gene sequences (N1 and N2) and the human RnaseP (RP) gene was used for the SAR-CoV-2 viral detection. The RT-qPCR primers and probes sequences were based on the CDC assay[73] with alternative fluorophores on the 5' end of each probe (along with 3' black hole quencher) for multiplexing. The N1 probe was labeled with ABY dye, the N2 probe with the FAM dye, and the RP probe with the VIC dye. The 16 μL reaction mix consists of 1× TaqPath 1-Step Multiplex Master Mix (Thermo Fisher Scientific), 250 nM of each forward and reverse primers for the N1 and N2 targets, 100 nM of each forward and reverse primer for the RP gene target, 62.5 nM for each of the N1 and N2 probes, 50 nM of the RP probe and 5 μl of sample RNA or controls. The RT-qPCR amplification was performed with one cycle at 53 °C for 10 mins and 95 °C for 2 min, followed by 40 cycles of 95 °C for 3 s and 60 °C for 30 s on a QuantStudio 7 Pro Real-Time PCR System (Thermo Fisher Scientific). The data was analyzed in the Design and Analysis 2.6.0 software (Thermo Fisher Scientific) using the auto baseline and threshold settings at 0.15 for N1 and N2 and 0.1 for the RP. Samples with amplification (Ct < 40) in both the N1 and N2 targets were determined as positive for SARS-CoV-2, according to the instructions for use[73]. In contrast, samples with amplification in only one of the targets were determined as inconclusive, and samples without amplification in both N1 and N2 targets were deemed negative for SARS-CoV-2. Each run included a negative extraction control using a pool of previously identified SARS-CoV-2 negative samples, positive extraction control, negative template control, and positive amplification control plasmid. All controls had to exhibit the expected performance for the assay to be considered valid. The RP gene was utilized for the analysis of human nasal swab samples performed at the same laboratory. It was not factored in for the result criteria for the air filter samples due to the low and inconsistent level of human cellular material trapped by the air filters.

### TrueMark respiratory panel
AerosolSense cartridges collected from community testing sites from October 25, 2021, to February 9, 2022, were tested for the presence of 40 different respiratory tract viral, bacterial, and fungal nucleic acids using the TrueMark Respiratory Panel 2.0 TaqMan Array Card (TAC) (Thermo Fisher Scientific). Substrates were extracted from the AerosolSense cartridge using sterile forceps, submerged into tubes containing 500 μL of PBS, vortexed for 5 s, and stored at 4 C for 10–30 min. Samples were removed from 4C and sterile forceps were used to disrupt the substrate by pressing it against the bottom of the tube several times to ensure bound particles were eluted into the PBS. According to the manufacturer's recommendations, the substrate was removed from the tube, and nucleic acids were isolated from the eluate using the Maxwell Viral Total Nucleic Acid Purification Kit (Promega) with the Maxwell 16 instrument (Promega). Briefly, 300 μL of the eluate was transferred to a tube containing 300 μL of lysis buffer, and 30 μL of Proteinase K. A nuclease-free water control was processed with each Maxwell run and used in the TrueMark protocol as a no-template control. Tubes were vortexed for 5 s and incubated on a heat block at 56 °C for 10 min. Following incubation, samples were centrifuged for 1 min to pellet any debris. Then 630 μL of each reaction mix was transferred into a Maxwell 16 cartridge, loaded into a Maxwell 16 instrument, and processed with the Viral Total Nucleic Acid program. Nucleic acids were eluted in 50 μL of RNase-free water. To perform the preamplification protocol, 5 μL of isolated nucleic acids were transferred into a PCR strip tube containing 2.5 μL of TaqPath 1-Step RT-qPCR Master Mix, CG (Thermo Fisher Scientific), and 2.5 μL of TrueMark Respiratory Panel 2.0 PreAmp Primers

(Thermo Fisher Scientific). Pre-amplification was performed on a thermocycler with the following cycling conditions: UNG incubation step at 25 °C for 2 min, reverse transcription at 50 °C for 30 min, UNG inactivation at 95 °C for 2 min, 14 cycles at 95 °C for 15 s (denaturation), 60 °C for 2 min (annealing and extension), followed by inactivation at 99.9 °C for 10 min, and 4 °C until samples were ready for use. Pre-amplified products were diluted 1:20 in nuclease-free water, and the TrueMark Respiratory Panel 2.0 Amplification Control (Thermo Fisher Scientific) was diluted 1:2 to include with every set of samples. TrueMark reaction mix was prepared by combining 20 µL of each diluted preamplified product with 50 µL of TaqMan Fast Advanced Master Mix (Thermo Fisher Scientific) and 30 µL of nuclease-free water. TAC were equilibrated to room temperature, and 100 µL of each reaction mix was loaded into its respective TAC port. TAC were centrifuged twice at 301 × *g* for 1 min each spin. TAC were sealed with a TAC Sealer, loaded into the Quant-Studio 7 Pro Real-Time PCR System (QS7), and run with the settings recommended by the manufacturer. Data were exported from the QS7 into the Thermo Fisher Design and Analysis Software 2.6.0. Data were analyzed according to the manufacturer's recommendations using the relative quantification module with the cycle relative threshold algorithm (Crt). Results were exported from the quality check module. Analysis was performed using a custom R script (v. 3.6.0) in RStudio (v. 1.3.959) to filter amplified results using the following cut-off values: amplification score >1.2 and Crt confidence >0.7. Samples were further filtered on reaction-specific Crt cut-off values determined in a LOD experiment (Supplementary Table 1). The TrueMark Respiratory Panel 2.0 includes technical control assays for human RNase P (RPPH1) and human 18S ribosomal RNA.

## TrueMark respiratory panel limit-of-detection estimation using contrived air samples

Contrived air samples were prepared using the TrueMark Respiratory Panel 2.0 Amplification Control (Thermo Fisher Scientific) to estimate the limit of detection (LOD) of the TrueMark Respiratory Panel 2.0 TaqMan Array Card. Briefly, two air cartridges were collected in an office for 48 h each. The air cartridge substrates were processed, and total nucleic acids were isolated as described above. Eluates from the four substrates were pooled together and aliquoted into five tubes. TrueMark amplification control plasmid, initially diluted in nuclease-free water, was added to four tubes at dilutions of 50 copies/µL, 10 copies/µL, 2.5 copies/µL, and 0.25 copies/µL. Final template concentrations for the preamplification reaction were 250, 50, 12.5, and 1.25 copies per reaction, respectively. No amplification control plasmid was added to the fifth tube that was used to determine the targets present in the background of contrived air samples collected from the empty office. An unused air cartridge was processed with the air samples as a negative template control. Four replicates of each contrived sample and control were processed through the reverse transcription, pre-amplification, dilution, and PCR protocols as described above. Data were analyzed in Thermo Fisher Design and Analysis Software 2.6.0 according to the manufacturer's recommendations. Replicates were called positive using the following cut-off values: amplification score >1.2 and Crt confidence >0.7. Cycle relative threshold (Crt) cut-off values were determined by averaging the Crt values of positive replicates at the lowest dilution concentration with at least 75% positive replicates. Any reaction targets that were detected in the contrived air sample or not detected at the highest amplification control dilution were excluded from the analysis and Crt cut-off values defaulted to the manufacturer's recommendation of Crt > 30.

### University of Minnesota

## Nucleic acid extraction and RT-qPCR
To elute the sample from the AerosolSense cartridges, both substrates were placed into 1 mL of PBS, making sure the substrates were fully saturated with PBS. A pipette was used to push down on the substrates

to extract as much eluate out of them as possible. Eluate was then transferred to a new tube. Samples were extracted using the *Quick*-RNA Viral Kit (Zymo Research). The extraction method followed manufacturer-recommended protocols with the notable exceptions of using 100 µL of starting material and eluting with 65 µL of appropriate elution material as indicated by manufacturer protocols. RT-qPCR reactions were set up in a 96-well Barcoded plate (Thermo Fisher Scientific) for either the N1 or N2 primers and probes with CDC-recommended sequences[74]. Then 5 µL extracted RNA was added to 15 µL qPCR master mix comprised of the following components: 8.5 µL nuclease-free water, 5 µL TaqMan™ Fast Virus 1-Step Master Mix (Thermo Fisher Scientific), and 1.5 µL primer/probe sets for either N1 or N2 (IDT, Cat# 10006713). SARS-CoV-2 RNA Control was obtained from Twist Biosciences (Genbank Ref. No. MN908947.3) and used as a positive control in each run. Reactions were cycled in a QuantStudio QS3 (Thermo Fisher Scientific) for one cycle of 50 °C for 5 min, followed by one cycle of 95 °C for 20 s, followed by 50 cycles of 95 °C for 3 s and 55 °C for 30 s. A minimum of two no-template controls were included on all runs. Baselines were allowed to calculate automatically, and a ΔRn threshold of 0.5 was selected and set uniformly for all runs. Ct values were exported and analyzed in Microsoft Excel. Amplification curves were manually reviewed. Samples with Ct < 40 in both N1 and N2 reactions were determined as positive for SARS-CoV-2. In contrast, samples with Ct < 40 in only N1 or N2 targets were determined as inconclusive results, and samples without amplification in both N1 and N2 targets were deemed negative for SARS-CoV-2.

## SARS-CoV-2 spike receptor binding domain sequencing
Targeted sequencing of the SARS-CoV-2 spike receptor-binding domain (RBD) was performed as previously described[27]. Primers used to amplify the spike RBD region for Illumina Miseq sequencing are shown in Supplementary Table 3. The data were analyzed using a custom workflow implemented in Snakemake and is publicly available at https://github.com/dholab/SARS-CoV-2-Spike-RBD-Analysis[75]. Briefly, paired-end reads were interleaved and merged into synthetic reads spanning the entire RBD PCR amplicon using bbmerge.sh (v38.93) from the bbtools package (sourceforge.net/projects/bbmap/) with default parameters. The merged reads were mapped to the SARS-CoV-2 reference sequence (Genbank MN908947.3) using minimap2 (v2.24) with the "-ax sr" preset for short reads. The resulting mapping file was sorted with samtools (v1.14). Reads that fully contain the desired amplicon sequence were extracted with the bedtools (v2.30.0) intersect tool. These reads were then downsampled to a target depth of 1000 reads using reformat.sh (v38.93) from the bbtools package. These downsampled reads were remapped to the MN908947.3 reference with minimap2. Residual PCR primer sequences were then trimmed with samtools ampliconclip using the "-hard-clip -both-ends" parameters. Next, a consensus sequence was generated by first generating a pileup with the samtools mpileup tool using default settings and then generating a consensus with ivar (v1.3.1) using the parameters "-q 20 -t 0 -m 20".

At the same time a consensus sequence was generated, the primer-trimmed reads were deduplicated to determine how many of the reads were identical, essentially defining pseudo-haplotypes. Vsearch (v2.21.1) fastx_uniques tool was used for deduplicating and enumerating the number of identical reads in each sample.

Lineage-defining mutations in the RBD were used to differentiate Delta from Omicron consensus sequences. Only one sample had evidence of mixed Delta and Omicron sequences.

### City of Milwaukee Health Department

## Nucleic acid extraction and RT-qPCR
Upon receipt of Thermo Scientific AerosolSense 2900 air sampler cartridge at City of Milwaukee Health Department Laboratory, collection

substrates were aseptically removed and transferred to a 5 mL sterile screw-cap tube filled with 1 mL of Remel viral transport medium. Samples were kept frozen at −70 °C until total nucleic extraction was performed using 200 μL elute and Applied Biosystems™ MagMAX™ Viral/Pathogen Nucleic Acid Isolation Kit using ThermoFisher Scientific KingFisher Flex instrument. Real-time RT-PCR setup was performed using 10 μL of extract and approved Applied Biosystems TaqPath™ COVID-19 Combo Kit containing three primer/probe sets specific to different SARS-CoV-2 genomic regions (open reading frame 1ab (ORF1ab), spike (S) protein, and nucleocapsid (N) protein-encoding genes) and primers/probes for bacteriophage MS2 which served as internal process control for nucleic acid extraction. RT-PCR assay was performed on Applied Biosystems 7500 Fast Dx Real-Time PCR System according to the TaqPath™ COVID-19 Combo Kit protocol. PCR results were interpreted using the Applied Biosystems COVID-19 Interpretive Software. A Ct cut-off value of ≤37 was used to call ORF1ab, S, and N targets as positive for the detection of SARS-CoV-2 viral RNA per the manufacturer's instructions for use. Air samples with amplification in two or more viral targets were reported as positive. Alternatively, samples with amplification in only one viral target were reported as inconclusive, and samples without amplification in any viral targets were deemed negative for SARS-CoV-2 viral RNA.

## SARS-CoV-2 sequencing

Samples with Ct values below 32 were sequenced using the ARTIC protocol and the Illumina DNA Prep library kit on a MiSeq instrument (https://www.protocols.io/view/sars-cov-2-sequencing-on-illumina-miseq-using-arti-bssjnecn). Data generated using the Integrated DNA Technologies ARCTIC V4 primer panel were analyzed using the Illumina® DRAGEN COVID Lineage App, which uses a customized version of the DRAGEN DNA pipeline to perform Kmer-based detection of SARS-CoV-2. The app aligns reads to a reference genome, calls variants, and generates a consensus genome sequence. Lineage/clade assignments were also confirmed using NextClade (https://clades.nextstrain.org/, version 1.14.0) and Pangolin COVID-19 Lineage Assigner (https://pangolin.cog-uk.io/, version 3.1.20) by uploading obtained FASTA files[76,77]. Consensus sequences generated and related metadata for environmental samples were shared publicly on Global Initiative on Sharing All Influenza Data (GISAID) (www.gisaid.org) (EPI_ISL_8879388 and EPI_ISL_8879389), the principal repository for SARS-CoV-2 genetic information.

### Mayo Clinic

## Nucleic acid extraction and RT-qPCR

Upon removal of the screw cap from the air sample cartridge in the biosafety level 2 cabinet, the air cartridge substrate was removed with a pair of disposable sterile forceps and transferred into a sterile tube containing 1 mL of PBS. The tube was vortexed for 10 s, and 200 μL of the sample was used for nucleic acid extraction and purification on the KingFisher Flex magnetic particle processor (Thermo Fisher Scientific) using the MagMAX Viral/Pathogen II Nucleic Acid Isolation Kit (Thermo Fisher Scientific) and MVP_Flex_200ul software program, with an elution volume of 50 uL. SARS-CoV-2 sequence targets (ORF1ab, N, and S gene sequences) were amplified and detected with the FDA-authorized TaqPath COVID-19 Combo Kit (Life Technologies Corp., Pleasanton, CA) on the Applied Biosystems 7500 Fast Dx Real-Time PCR System (Life Technologies Corp.) per assay manufacturer's instructions for use. A Ct cut-off value of ≤37 was used to call ORF1ab, S, and N targets as positive for the detection of SARS-CoV-2 viral RNA per the manufacturer's instructions for use. Air samples with amplification in two or more viral targets were reported as positive. Alternatively, samples with amplification in only one viral target were reported as negative, and samples without amplification in any viral targets were deemed negative for SARS-CoV-2 viral RNA.

### Reporting summary

Further information on research design is available in the Nature Research Reporting Summary linked to this article.

## Data availability

The SARS-CoV-2 sequencing data generated in this study have been deposited in the Sequence Read Archive (SRA) under bioprojects PRJNA811594 and PRJNA856293. Consensus sequences generated from environmental samples with ARCTIC V4 were shared publicly on Global Initiative on Sharing All Influenza Data (GISAID) (www.gisaid.org) (EPI_ISL_8879388 and EPI_ISL_8879389). Air sample metadata and SARS-CoV-2 RT-qPCR are provided in Supplementary Data 1. The TrueMark air sample data generated in this study are provided in Supplementary Data 2. We obtained county-wide COVID-19 case data for Dane County from Public Health Madison and Dane County COVID-19 Dashboard (https://publichealthmdc.com/coronavirus/dashboard).

## Code availability

Code to replicate SARS-CoV-2 sequencing analysis are available at https://github.com/dholab/SARS-CoV-2-Spike-RBD-Analysis.

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

## Acknowledgements

This work was made possible by financial support through Rockefeller Regional Accelerator for Genomic Surveillance (133 AAJ4558 D.H.O., S.L.O., and T.C.F.), Wisconsin Department of Health Services Epidemiology and Laboratory Capacity funds (144 AAJ8216 D.H.O. and T.C.F.), and National Institutes of Health grant (U01DA053893 M.C.J.). M.D.R. is supported by the National Institute of Allergy and Infectious Diseases of the National Institutes of Health under Award Number T32AI55397. We would like to thank all of the participating congregate settings for their partnership during this study. We would like to thank Thermo Fisher Scientific, Inc. for donating 4 of the AerosolSense samplers used in the data described in this work. We would like to acknowledge Elizabeth Durkes and Hannah Kraussel, Public Health Emergency Response Planning Coordinators, Lindor Schmidt, Environmental & Disease Control Specialist, and Dr. Nicholas Tomaro, Emergency Preparedness Environmental Health Director at the City of Milwaukee Health Department for their assistance in coordinating site selection, sampling, cartridge replacement, and unit decontamination processes. We would also like to acknowledge Dr. Arun Ramaiah, Bioinformatician at MHD lab, for depositing air sample sequencing data in SRA.

## Author contributions

M.D.R contributed to the conceptualization, data curation, formal analysis, investigation, methodology, project administration, visualization, writing—original draft preparation, writing—review and editing. C.M.N. contributed to data curation, investigation, methodology, supervision, writing—review and editing. S.F.B. contributed to project administration, data curation, resources, visualization, writing—review and editing. M.R.S. contributed to the investigation, data curation, methodology, writing—review and editing. R.W.W. contributed to data curation, methodology, writing—review and editing. A.K.Y. contributed to resources, writing—review and editing. E.J.O. contributed to software, data curation, writing—review and editing. N.D. contributed to the investigation, data curation, methodology, writing—review and editing. A.L. contributed to the investigation, formal analysis, data curation, methodology, supervision, writing—review and editing. K.P.P. contributed to resources, supervision, writing—review and editing. N.S. contributed to resources, supervision, data curation, writing—review and editing. J.A.M. contributed to resources, supervision, data curation, writing—review and editing. M.A.A. contributed to resources, supervision, data curation, writing—review and editing. W.M.R contributed to resources, supervision, data curation, writing—review and editing. J.A.Z. contributed to the investigation, developed methods, sample processing, performed analysis, data curation, writing—review and editing. M.K. Contributed to the method development, analysis, data curation, writing—review and editing. L.J.B. contributed to conceptualization, data curation, formal analysis, investigation, methodology, writing—review and editing. E.C.B. contributed to the investigation, formal analysis, data curation, methodology, writing—review and editing. D.C. contributed to the investigation, formal analysis, data curation, methodology, writing—review and editing. C.R. contributed to the investigation, data curation, methodology, writing—review and editing. D.A.G contributed to formal analysis, data curation, writing—review and editing. J.D.Y. contributed to conceptualization, supervision, methodology, writing—review and editing. S.B. contributed to resources, conceptualization, supervision, writing—review and editing. M.C.J. contributed to the investigation, data curation, methodology, writing—review and editing. M.T.A. contributed to conceptualization, supervision, methodology, writing—review and editing. T.C.F. contributed to the

conceptualization, funding acquisition, writing—review and editing. D.H.O. contributed to the conceptualization, formal analysis, software, funding acquisition, methodology, supervision, project administration, writing—original draft preparation, writing—review and editing. S.L.O. contributed to the conceptualization, funding acquisition, methodology, supervision, project administration, writing—review and editing.

## Competing interests

The authors declare no competing interests.
