## [Peer Review File · Nature Communications]

SARS-CoV-2 and other respiratory pathogens are detected in continuous air samples from congregate settingsREVIEWER COMMENTS

Reviewer #1 (Remarks to the Author):

This manuscript describes a large collection of air samples from real-world settings in Wisconsin and Minnesota in late 2021 and early 2022, that were analyzed for SARS-CoV-2 and other respiratory pathogens. Sampling locations included a hospital, cafeteria, brewery taproom, bar, schools, shelters, and campus facilities. A mask mandate was in place during most of the study, except where all people were known to be vaccinated. Out of 527 samples, 20% were positive for viral RNA and 10% were inconclusive. Positive samples occurred at 14/15 sites, even when risk mitigation strategies were in place.

This is one of the largest, if not the largest, collection of air samples analyzed for SARS-CoV-2 that I have seen. The positivity rate of 20% is striking; on average, viral RNA was present in the air in one out of five public indoor spaces. It is also interesting that the researchers detected, not infrequently, a menagerie of other pathogens, including adenovirus, a seasonal coronavirus, influenza virus, RSV, VZV, Epstein-Barr, K. pneumoniae, and more. Sequencing revealed mainly Delta and Omicron variants, along with some unexpected variants.

The authors estimated the cost of daily air surveillance in schools in Wisconsin and showed that it would have been similar to the amount spent on testing of individuals. The authors also provide a useful perspective on how to interpret and share air sampling data, as it may not be in the building manager's best interest to disclose such information.

The Introduction cites the relevant literature efficiently. The presentation of the material is clear enough but not at the level of concision, precision, and crispness I would expect for a paper in this journal. Some interpretation and commentary appear in the Results section, while results about the temporal pattern of samples in the brewery taproom appear in the Discussion section. Specific comments are listed below.

1. line 34: "detected 106 SARS-CoV-2 positive samples" The abstract should explain what is meant by "positive," in this case positive for viral RNA.
2. line 168: "We were able to intermittently identify SARS-CoV-2 positive air cartridges at 14 of the 15 surveillance sites, even when intensive risk mitigation strategies were implemented by public health (Table 1)." The placement of the label for Table 1 implies that it includes information about risk mitigation strategies. As it does not, I suggest moving it to appear after the word "sites". Also, strategies are not implemented by public health.
3. line 172: "By testing air samplers in these real-world settings, it was impossible..." This paragraph requires corrections to spelling and grammar (e.g., dangling participle and more). The epidemiological details about who tested positive on what days and who worked with whom detract from the main message. I suggest condensing this text to focus simply on the timing of the cases compared to the timing of the positive air samples, while emphasizing the delay between actual infection and confirmation of a case.
4. line 197: The utility of week-long samples seems limited because by the time results are available, exposure and transmission might have occurred days earlier.
5. line 334: "Quantitative RT-PCR assays of nucleic acids in a test tube provide semi-quantitative information on the viral load present in ambient air, but these viral load data should be interpreted with care." Actually, qRT-PCR can provide excellent quantitative information on the viral load present in ambient air, particularly if the assay included a calibration curve to convert Ct values to gene copies. If the particle collection efficiency of the cartridges is near 100%, then you can divide the number of gene copies detected by the total volume of air sampled to derive a concentration in air in terms of gene copies per cubic meter of air. You can calculate the total volume of air sampled by multiplying the volume flow rate of the sampler (200 L/min) by the number of minutes it ran. So if it ran for 24 hours, the total volume of air sampled would be $200 \text{ L/min} * 1440 \text{ min} = 288,000 \text{ L} = 288 \text{ m}^3$.
6. line 398: "Samplers were...calibrated to sample 200 liters of air per minute." How long did the samplers run (i.e., what was the total volume of air sampled)?
7. Figure 1: This figure, showing the workflow, can be moved to the supplementary information.

Linsey Marr

Reviewer #2 (Remarks to the Author):

Ramuta et al. presented a way of surveillance of SARS-CoV-2 and other respiratory pathogens in congregate settings using air sampling. This method shows its promise as complement to individual testing, wastewater surveillance etc., since the authors demonstrated its ability to detect SARS-CoV-2, including new VOCs early, monitor multiple pathogens simultaneously, and reveal hyperlocal trends of respiratory infectious disease community transmission. The paper is well written. However, I find that the paper has a few major issues, particularly on how well the detection of SARS-CoV-2 in air samples reflects the infection risk in the settings of interest. I would recommend its publication after they are addressed.

An assumption of relatively fast and good mixing of indoor air in congregate settings is needed for the air samples to reflect the presence of SARS-CoV-2 in the air faithfully. However, not all congregate settings of interest in this study meet this requirement. When the setting is large and/or has relatively poor air exchange between different parts, e.g., in schools, it would be difficult to detect the presence of an infected teacher or student who just passes by the air sampler in a high-traffic hallway and stays in a classroom far from it for a long period. But this individual poses significant transmission risk to other occupants in that room. More air samples for different air compartments in a single setting can address the heterogeneity issue, but will substantially increase the cost of the monitoring program.

Even if the air heterogeneity issue is ignored, it is unclear to me whether presence of infected people (and hence significant risk) translates to positive samples in that setting. I understand the air sampling in this study is not for quantitatively monitoring airborne infectious virus. But the presence of infected people and/or significant transmission risk should be detected qualitatively, so that the air sampling is effective. What nucleic acid material above the LOD means is not clearly explained in the paper, nor whether it corresponds to presence of infected people or vice versa. Only a single case study reported in the paragraph starting from Line 172 out of the large amount of monitoring data the authors have does not convince me that a qualitative relationship between positive samples and risk is established.

I suggest that the authors systematically analyze their monitoring data to address the two issues above: i) whether a single air sampler is sufficient for congregate settings with large/complex structure and/or limited indoor air exchange; ii) what nucleic acid material above the LOD really means in this monitoring program.

Specific comments:

Line 164: I do not fully understand why exceptions were made for places where all people were vaccinated. Vaccinated people with breakthrough infection still shed virus significantly, though with a reduced quantity. I do not expect very different air sampling results between two similar settings with 100% and 95% vaccination rates.

Lines 196-211: the point of air sampling for extended duration is not fully clear to me. I agree that sampling for a week can yield similar results as daily sampling, but the former is less informative than the latter. The generation times of Delta and Omicron VOCs are short. A week is enough for multiple generations of transmission. If the monitoring data cannot inform the management of a congregate setting of transmission risk in a timely manner, the monitoring is not highly effective.

Line 297: the paragraph from Line 297 compared the costs of individual testing and air sampling for monitoring COVID-19 in schools. I do not agree with the way of comparison described in the paper. Individual testing produces key information that who need to self-isolate while air sampling does not. If individual testing is only to know if virus is present in the setting, one PCR run for a sample with 10 or 20 swabs mixed, as used frequently in mass testing in China, can greatly reduce the cost and yield a similar level of information as air sampling.

Line 341: in settings with complex structure such as schools, CO2 level evaluation cannot necessarily help optimize the placement of air samplers. Different classrooms with a similar occupancy level should have a similar CO2 level that is generally higher than common areas such as a hallway. But an air sampler placed in a classroom does not provides much information of the

infection risk in another classroom.

Lines 344-354: this paragraph discussed a limitation of the air sampling method. The authors should note that it can cause serious biases for settings where UV germicidal irradiation is widely used, such as hospitals. UVGI inactivates virus without largely destroying genetic material.

Area of expertise

I am an indoor air researcher and do not feel qualified to comment on the details of respiratory pathogen detection techniques.

Reviewer #3 (Remarks to the Author):

This manuscript describes the results of a large air sampling deployment to sample for SARS-CoV-2 and other respiratory viruses. In general, I find this article to be informative and of high quality. I feel that with very minor adjustments, this can be ready for publication.

Comment 1. What is the maximum Ct accepted to declare a SARS-CoV-2 positive for the individual assays. I couldn't find that value given in the manuscript. There has been some discussion in the community around what constitutes an acceptable Ct value when processing environmental samples, so I think it is important to both clearly state what was done, and the rationale.

Comment 2. Following the comment above, it would be helpful to see Ct values in the supplement, or in a data repository. Since the paper reports only positive/negative/inconclusive, its not possible to recover any quantitative or semi-quantitative estimates. Making the Ct values available, would be helpful perspective.

Comment 3. There are a few literature references that are missing that I think should be included in the discussion.

Cultivation of SARS-CoV-2 from collected aerosol

Lednicky JA, Lauzardo M, Alam MM, Elbadry MA, Stephenson CJ, Gibson JC, et al. Isolation of SARS-CoV-2 from the air in a car driven by a COVID patient with mild illness. *Int J Infect Dis.* 2021;108:212–6. <https://doi.org/10.1016/j.ijid.2021.04.063>

Santarpia, J.L., Herrera, V.L., Rivera, D.N. et al. The size and culturability of patient-generated SARS-CoV-2 aerosol. *J Expo Sci Environ Epidemiol* (2021). <https://doi.org/10.1038/s41370-021-00376-8>

Study on sampling of SARS-CoV-2 aerosols

Ratnesar-Shumate S, Bohannon K, Williams G, Holland B, Krause M, Green B, et al. Comparison of the performance of aerosol sampling devices for measuring infectious SARS-CoV-2 aerosols. *Aerosol Sci Technol.* 2021. <https://doi.org/10.1080/02786826.2021.1910137>

Previous study around aerosol sampling in schools

Crowe J, Schnaubelt AT, SchmidtBonne S, et al. Assessment of a Program for SARS-CoV-2 Screening and Environmental Monitoring in an Urban Public School District. *JAMA Netw Open.* 2021;4(9):e2126447. doi:10.1001/jamanetworkopen.2021.26447

Reviewer #4 (Remarks to the Author):

Ramuta et al. report findings from an air sampling surveillance strategy for SARS-CoV-2 and other respiratory pathogens implemented in 15 congregate settings located in two US states. The approach utilized active air samplers and cartridges from ThermoFisher and was coupled with a standardized workflow to manage sampling data. The authors found that continuous air sampling was effective in routinely detecting viral nucleic acids from SARS-CoV-2 in congregate settings, as well as other respiratory pathogens (e.g. influenza A virus) that correlated with other epidemiological and diagnostic data. The authors conclude that continuous air sampling is an effective approach to conducting SARS-CoV-2 surveillance.

Overall, the study is comprehensive and very well-presented. The various amounts of data collected as part of this study allows for a multi-faceted and critical analysis regarding the utility of using an air surveillance strategy as part of a routine epidemiological surveillance approach for respiratory pathogen detection. This work is valuable to the scientific community, but more importantly the public health community where such strategies might better inform mitigation strategies. The authors presented well-tempered and evidenced based interpretations of their findings and discussed many of the known limitations of air sampling data. Below are some additional comments and suggestions the authors might consider to further enhance the quality of their manuscript.

General Comments

While there are different views among the scientific community on how best to communicate the pathogen positivity data from air sample collections, given that one of the limitations to using air sampling results for mitigation policies is the lack of viability data, I suggest that the authors make it more clear throughout the paper that what is being detected in the samples is SARS-CoV-2 RNA and not necessarily the whole viable virus. This was clearly noted for the influenza A virus results, but is not as consistent for the reported SARS-CoV-2 data.

While the direct sampling and workflow are standardized, the laboratory methods used across sites employ different extraction protocols and RT-qPCR platforms. As it pertains to comparing results across study sites, this is a potential limitation the authors should discuss more fully. What might be the variability of final viral RNA concentrations using different extraction methods? Additionally, what are the differences in the limits of detection (LODs) for the various testing platforms used? It is possible that that specifications for these kits and platforms result in a negligible variability, but this should be evaluated and clearly communicated by the authors.

Abstract

Line 38-40: Using the word 'alternative' here implies that air sampling should replace individual sampling in congregate settings. The authors seem to state in the discussion, I think more appropriately, that the study data shows air sampling to be an effective "additional" tool to other surveillance testing strategies. This should be clarified here in the abstract.

Introduction

Line 43-44: I suggest a different wording than "upon its arrival". Perhaps "detect the first wave of SARS-CoV-2 cases in the United States".

Line 50-51: Suggest removing "The explosive spread of Omicron was unprecedented".

Line 87: should this be 'but sample collection...'?

Methods

Line 398: convert feet to metric

Supplemental File

Supplemental Figure 02: correct to 'follow-up'

(Line numbers in the response match the merged PDF line numbers of the revised manuscript)

REVIEWER COMMENTS

We thank the reviewers for their perceptive comments and overall enthusiasm for our manuscript. We have responded point-by-point to each comment and believe the manuscript is much improved as a consequence.

Reviewer #1 (Remarks to the Author):

The presentation of the material is clear enough but not at the level of concision, precision, and crispness I would expect for a paper in this journal. Some interpretation and commentary appear in the Results section, while results about the temporal pattern of samples in the brewery taproom appear in the Discussion section. Specific comments are listed below.

Response: We have revised the Results and Discussion to improve the readability of the manuscript. Specifically, we moved the results about the vaccine mandate and temporal pattern of samples from the brewery taproom to the results section (lines: 276-282). We also removed several lines of commentary from the results section (lines: 177 and 193).

1. line 34: “detected 106 SARS-CoV-2 positive samples” The abstract should explain what is meant by “positive,” in this case, positive for viral RNA.

Response: We have clarified what a “positive” air sample result means in the abstract by stating that the samples were positive for viral RNA (line: 37).

2. line 168: “We were able to intermittently identify SARS-CoV-2 positive air cartridges at 14 of the 15 surveillance sites, even when intensive risk mitigation strategies were implemented by public health (Table 1).” The placement of the label for Table 1 implies that it includes information about risk mitigation strategies. As it does not, I suggest moving it to appear after the word “sites”. Also, strategies are not implemented by public health.

Response: We apologize for the confusion. We have changed the placement of Table 1 to appear after the word “sites” in line 176. We have also changed the sentence in line 178 to clarify that risk mitigation strategies are recommended but not implemented by public health.

3. line 172: “By testing air samplers in these real-world settings, it was impossible...” This paragraph requires corrections to spelling and grammar (e.g., dangling participle and more). The epidemiological details about who tested positive on what days and who worked with whom detract from the main message. I suggest condensing this text to focus simply on the timing of the cases compared to the timing of the positive air samples, while emphasizing the delay between actual infection and confirmation of a case.

Response: We apologize for the grammatical errors. We have changed the sentence to correct these errors (lines 179-181). We have also condensed this section of the results to focus less on the individual case details and more on the timing of cases compared to the timing of positive air samples (lines 183-188).

4. line 197: The utility of week-long samples seems limited because by the time results are available, exposure and transmission might have occurred days earlier.

Response: We agree there are limitations to weekly testing. Testing more frequently is optimal for using air sampling data to quickly implement risk mitigation strategies to prevent further transmission in a congregate setting. However, daily testing is expensive, and the initial feedback we received during our early daily testing was that this would be too expensive for most sites. Weekly sampling is more cost-effective and can still provide useful information about the relative risk of virus presence and track variants of interest. We have expanded the discussion to highlight the limitations and benefits of weekly and more frequent sampling intervals (lines: 329-340).

5. line 334: “Quantitative RT-PCR assays of nucleic acids in a test tube provide semi-quantitative information on the viral load present in ambient air, but these viral load data should be interpreted with care.” Actually, qRT-PCR can provide excellent quantitative information on the viral load present in ambient air, particularly if the assay included a calibration curve to convert Ct values to gene copies. If the particle collection efficiency of the cartridges is near 100%, then you can divide the number of gene copies detected by the total volume of air sampled to derive a concentration in air in terms of gene copies per cubic meter of air. You can calculate the total volume of air sampled by multiplying the volume flow rate of the sampler (200 L/min) by the number of minutes it ran. So if it ran for 24 hours, the total volume of air sampled would be $200 \text{ L/min} * 1440 \text{ min} = 288,000 \text{ L} = 288 \text{ m}^3$.

Response: This is an excellent point. Unfortunately, we did not perform absolute quantification of SARS-CoV-2 RNA. We now note in the discussion that this would be a useful addition to generate quantitative information on air samples when testing with qRT-PCR (lines: 360-364).

6. line 398: “Samplers were...calibrated to sample 200 liters of air per minute.” How long did the samplers run (i.e., what was the total volume of air sampled)?

Response: We have precise timestamps for each sample. We used this information to determine the amount of time that each air sample was run and calculated the total volume of air sampled for each sample in Supplementary Data 1.

7. Figure 1: This figure, showing the workflow, can be moved to the supplementary information.

Response: We have moved the workflow figure into the supplementary information (Supplementary Figure 1).

Reviewer #2 (Remarks to the Author):

Ramuta et al. presented a way of surveillance of SARS-CoV-2 and other respiratory pathogens in congregate settings using air sampling. This method shows its promise as complement to individual testing, wastewater surveillance etc., since the authors demonstrated its ability to detect SARS-CoV-2, including new VOCs early, monitor multiple pathogens simultaneously, and reveal hyperlocal trends of respiratory infectious disease community transmission. The paper is well written. However, I find that the paper has a few major issues, particularly on how well the detection of SARS-CoV-2 in air samples reflects the infection risk in the settings of interest. I would recommend its publication after they are addressed.

An assumption of relatively fast and good mixing of indoor air in congregate settings is needed for the air samples to reflect the presence of SARS-CoV-2 in the air faithfully. However, not all congregate settings of interest in this study meet this requirement. When the setting is large and/or has relatively poor air exchange between different parts, e.g., in schools, it would be difficult to detect the presence of an infected teacher or student who just passes by the air sampler in a high-traffic hallway and stays in a classroom far from it for a long period. But this individual poses significant transmission risk to other occupants in that room. More air samples for different air compartments in a single setting can address the heterogeneity issue, but will substantially increase the cost of the monitoring program.

Response: The reviewer makes an excellent point about the challenges associated with establishing air monitoring programs in congregate settings with complex structure, such as schools. We have mentioned these limitations in the discussion in the revised manuscript and have expanded the Discussion to note that the use of multiple air samplers or other sampling strategies may be required for better surveillance in these settings (lines 364-375).

Even if the air heterogeneity issue is ignored, it is unclear to me whether presence of infected people (and hence significant risk) translates to positive samples in that setting. I understand the air sampling in this study is not for quantitatively monitoring airborne infectious virus. But the presence of infected people and/or significant transmission risk should be detected qualitatively, so that the air sampling is effective. What nucleic acid material above the LOD means is not clearly explained in the paper, nor whether it corresponds to presence of infected people or vice versa. Only a single case study reported in the paragraph starting from Line 172 out of the large amount of monitoring data the authors have does not convince me that a qualitative relationship between positive samples and risk is established.

Response: The reviewer poses an important question that we now bring up as a potential future study in the discussion (lines: 396-410). This requires partnerships with congregate settings that have well-established, intensive surveillance testing programs to compare air sampling data with case reports. We recently established one such partnership with the ORCHARDS program

(<https://www.fammed.wisc.edu/orchards/>) and hope to address this issue systematically in the 2022-2023 academic year.

I suggest that the authors systematically analyze their monitoring data to address the two issues above: i) whether a single air sampler is sufficient for congregate settings with large/complex structure and/or limited indoor air exchange; ii) what nucleic acid material above the LOD really means in this monitoring program.

Response: We agree that the utility of a single air sampler diminishes as the size and complexity of a structure increases and now note this in line 364-369. While we cannot say with certainty that detection of RNA above the LOD correlates with the presence of infectious virus, it is nonetheless striking that the lowest levels of RNA were observed in the summer of 2021 when community transmission was lowest and highest in early 2022 when the Omicron-fueled community transmission were highest.

Specific comments:

1. Line 164: I do not fully understand why exceptions were made for places where all people were vaccinated. Vaccinated people with breakthrough infection still shed virus significantly, though with a reduced quantity. I do not expect very different air sampling results between two similar settings with 100% and 95% vaccination rates.

Response: We agree with the reviewer's comment that vaccinated people with breakthrough infections can still shed infectious virus and pose a significant risk to others (indeed, we were among the first to raise this alarm in July 2021). In line 164 (now lines:167-171) we are describing the emergency mask mandate issued by Public Health Madison and Dane County during the period of our study.

2. Lines 196-211: the point of air sampling for extended duration is not fully clear to me. I agree that sampling for a week can yield similar results as daily sampling, but the former is less informative than the latter. The generation times of Delta and Omicron VOCs are short. A week is enough for multiple generations of transmission. If the monitoring data cannot inform the management of a congregate setting of transmission risk in a timely manner, the monitoring is not highly effective.

Response: The reviewer poses an important question about the utility of weekly sampling. We have expanded the discussion to highlight the limitations and benefits of weekly sampling intervals in the context of the management of a congregate setting and public health (lines: 329-340). There is a real tension between different sampling intervals (see above response to Reviewer 1, who raised the same issue). Future work will need to be done to construct individualized air sampling workflows that are both actionable and cost-effective for each specific setting.

3. Line 297: the paragraph from Line 297 compared the costs of individual testing and air sampling for monitoring COVID-19 in schools. I do not agree with the way of comparison

described in the paper. Individual testing produces key information who need to self-isolate while air sampling does not. If individual testing is only to know if virus is present in the setting, one PCR run for a sample with 10 or 20 swabs mixed, as used frequently in mass testing in China, can greatly reduce the cost and yield a similar level of information as air sampling.

Response: Per the reviewer's request, we have limited the economic discussion on the cost of an air sampling program and have removed comparisons to individual testing program costs (lines 316-327).

4. Line 341: in settings with a complex structure such as schools, CO₂ level evaluation cannot necessarily help optimize the placement of air samplers. Different classrooms with a similar occupancy level should have a similar CO₂ level that is generally higher than common areas such as a hallway. But an air sampler placed in a classroom does not provide much information of the infection risk in another classroom.

Response: The reviewer brings up a valid point about the limitations of using CO₂ monitoring to help optimize the placement of air samplers in settings with complex structures. We have omitted this sentence from the discussion.

5. Lines 344-354: this paragraph discussed a limitation of the air sampling method. The authors should note that it can cause serious biases in settings where UV germicidal irradiation is widely used, such as hospitals. UVGI inactivates viruses without largely destroying genetic material.

Response: We have added this caveat in the discussion (lines: 378-385).

Reviewer #3 (Remarks to the Author):

This manuscript describes the results of a large air sampling deployment to sample for SARS-CoV-2 and other respiratory viruses. In general, I find this article to be informative and of high quality. I feel that with very minor adjustments, this can be ready for publication.

1. What is the maximum Ct accepted to declare a SARS-CoV-2 positive for the individual assays. I couldn't find that value given in the manuscript. There has been some discussion in the community around what constitutes an acceptable Ct value when processing environmental samples, so I think it is important to both clearly state what was done, and the rationale.

Response: We have clarified the Ct cut-off values used for each assay used in the study in their respective methods sections and included Ct cut-off values in Supplementary data 1. As noted by Reviewers 1 and 2, there is a tension between sampling duration, turnaround time, and ability to respond to the results. To minimize turnaround time after sampling, each partnering site used its own SARS-CoV-2 PCR assay. We now describe these assays more fully (lines 474-476; 496-505; 539-546; Supplementary Table 1; 588-591; 631-636; 662-667) and note that the Ct values are not necessarily directly comparable between sites (lines 149-152).

2. Following the comment above, it would be helpful to see Ct values in the supplement, or in a data repository. Since the paper reports only positive/negative/inconclusive, its not possible to recover any quantitative or semi-quantitative estimates. Making the Ct values available, would be helpful perspective.

Response: We have added the SARS-CoV-2 Ct values for every sample in Supplementary Data 1.

3. There are a few literature references that are missing that I think should be included in the discussion.

Response: We thank the reviewer for pointing these references out and have incorporated them into the discussion (lines: 371; 388; 390).

Cultivation of SARS-CoV-2 from collected aerosol

Lednicky JA, Lauzardo M, Alam MM, Elbadry MA, Stephenson CJ, Gibson JC, et al. Isolation of SARS-CoV-2 from the air in a car driven by a COVID patient with mild illness. *Int J Infect Dis.* 2021;108:212–6. <https://doi.org/10.1016/j.ijid.2021.04.063>

Santarpia, J.L., Herrera, V.L., Rivera, D.N. et al. The size and culturability of patient-generated SARS-CoV-2 aerosol. *J Expo Sci Environ Epidemiol* (2021). <https://doi.org/10.1038/s41370-021-00376-8>

Study on sampling of SARS-CoV-2 aerosols

Ratnesar-Shumate S, Bohannon K, Williams G, Holland B, Krause M, Green B, et al. Comparison of the performance of aerosol sampling devices for measuring infectious SARS-CoV-2 aerosols. *Aerosol Sci Technol.* 2021. <https://doi.org/10.1080/02786826.2021.1910137>

Previous study around aerosol sampling in schools

Crowe J, Schnaubelt AT, SchmidtBonne S, et al. Assessment of a Program for SARS-CoV-2 Screening and Environmental Monitoring in an Urban Public School District. *JAMA Netw Open.* 2021;4(9):e2126447. doi:10.1001/jamanetworkopen.2021.26447

Reviewer #4 (Remarks to the Author):

Ramuta et al. report findings from an air sampling surveillance strategy for SARS-CoV-2 and other respiratory pathogens implemented in 15 congregate settings located in two US states. The approach utilized active air samplers and cartridges from ThermoFisher and was coupled with a standardized workflow to manage sampling data. The authors found that continuous air

sampling was effective in routinely detecting viral nucleic acids from SARS-CoV-2 in congregate settings, as well as other respiratory pathogens (e.g. influenza A virus) that correlated with other epidemiological and diagnostic data. The authors conclude that continuous air sampling is an effective approach to conducting SARS-CoV-2 surveillance.

Overall, the study is comprehensive and very well-presented. The various amounts of data collected as part of this study allows for a multi-faceted and critical analysis regarding the utility of using an air surveillance strategy as part of a routine epidemiological surveillance approach for respiratory pathogen detection. This work is valuable to the scientific community, but more importantly the public health community where such strategies might better inform mitigation strategies. The authors presented well-tempered and evidenced based interpretations of their findings and discussed many of the known limitations of air sampling data. Below are some additional comments and suggestions the authors might consider to further enhance the quality of their manuscript.

General Comments

1. While there are different views among the scientific community on how best to communicate the pathogen positivity data from air sample collections, given that one of the limitations to using air sampling results for mitigation policies is the lack of viability data, I suggest that the authors make it more clear throughout the paper that what is being detected in the samples is SARS-CoV-2 RNA and not necessarily the whole viable virus. This was clearly noted for the influenza A virus results but is not as consistent for the reported SARS-CoV-2 data.

Response: We have edited the text to clarify that we were analyzing samples to detect pathogen genetic material and not viable virus, using the term 'viral RNA' or 'SARS-CoV-2 RNA' throughout. We also now note in the Discussion that assessing the viability of virus captured by air samples is an important question to address in the future (lines 377-394). Indeed, a related issue was raised by another reviewer who notes that in the presence of UV germicidal irradiation an air sample could be positive for viral RNA but not contain infectious virus.

2. While the direct sampling and workflow are standardized, the laboratory methods used across sites employ different extraction protocols and RT-qPCR platforms. As it pertains to comparing results across study sites, this is a potential limitation the authors should discuss more fully. What might be the variability of final viral RNA concentrations using different extraction methods? Additionally, what are the differences in the limits of detection (LODs) for the various testing platforms used? It is possible that the specifications for these kits and platforms result in a negligible variability, but this should be evaluated and clearly communicated by the authors.

Response: The reviewer makes an excellent point that was also noted by Reviewer 3 (see response above). We now more clearly describe the different methods used for RT-qPCR and

caution against directly comparing Ct values between assays (lines 149-152; 474-476; 496-505; 539-546; Supplementary Table 1; 588-591; 631-636; 662-667).

3. Line 38-40: Using the word 'alternative' here implies that air sampling should replace individual sampling in congregate settings. The authors seem to state in the discussion, I think more appropriately, that the study data shows air sampling to be an effective "additional" tool to other surveillance testing strategies. This should be clarified here in the abstract.

Response: The reviewer makes an excellent point. We have changed the sentence to highlight that air sampling is an effective tool to be used in conjunction with other surveillance methods instead of replacing them (lines: 42-43).

4. Line 43-44: I suggest a different wording than "upon its arrival". Perhaps "detect the first wave of SARS-CoV-2 cases in the United States".

Response: We have omitted "upon its arrival" from the sentence and added the reviewer's suggestion of "detect the first wave of SARS-CoV-2 cases in the United States" (line: 47).

5. Line 50-51: Suggest removing "The explosive spread of Omicron was unprecedented".

Response: We have removed this phrase from the sentence (line: 53).

6. Line 87: should this be 'but sample collection...'?

Response: We have changed the sentence to say "but sample collect" instead of "and sample collection" (line: 93).

7. Line 398: convert feet to metric

Response: We have converted feet to meters (lines: 449).

8. Supplemental Figure 02: correct to 'follow-up'

Response: We have modified supplemental figure 02 to reflect this correction.

REVIEWERS' COMMENTS

Reviewer #2 (Remarks to the Author):

If the authors provided a version of the revised manuscript with track changes, it would be much easier for me to examine their responses to the reviewers' comments. Nevertheless, this is only a suggestion for the peer-review process. In terms of the contents of the responses and the revised paper, I think that the authors have adequately addressed the comments and recommend publication of the revised paper.

Reviewer #3 (Remarks to the Author):

I feel that all of my comments have been addressed.